# ON THE MAPPING BETWEEN HOPFIELD NETWORKS AND RESTRICTED BOLTZMANN MACHINES

**Matthew Smart**
Department of Physics
University of Toronto
msmart@physics.utoronto.ca

**Anton Zilman**
Department of Physics
and Institute for Biomedical Engingeering
University of Toronto
zilmana@physics.utoronto.ca

## ABSTRACT

Hopfield networks (HNs) and Restricted Boltzmann Machines (RBMs) are two important models at the interface of statistical physics, machine learning, and neuroscience. Recently, there has been interest in the relationship between HNs and RBMs, due to their similarity under the statistical mechanics formalism. An exact mapping between HNs and RBMs has been previously noted for the special case of orthogonal ("uncorrelated") encoded patterns. We present here an exact mapping in the case of correlated pattern HNs, which are more broadly applicable to existing datasets. Specifically, we show that any HN with $N$ binary variables and $p < N$ arbitrary binary patterns can be transformed into an RBM with $N$ binary visible variables and $p$ gaussian hidden variables. We outline the conditions under which the reverse mapping exists, and conduct experiments on the MNIST dataset which suggest the mapping provides a useful initialization to the RBM weights. We discuss extensions, the potential importance of this correspondence for the training of RBMs, and for understanding the performance of deep architectures which utilize RBMs.

## 1 INTRODUCTION

Hopfield networks (HNs) (Hopfield, 1982; Amit, 1989) are a classical neural network architecture that can store prescribed patterns as fixed-point attractors of a dynamical system. In their standard formulation with binary valued units, HNs can be regarded as spin glasses with pairwise interactions $J_{ij}$ that are fully determined by the patterns to be encoded. HNs have been extensively studied in the statistical mechanics literature (e.g. (Kanter & Sompolinsky, 1987; Amit et al., 1985)), where they can be seen as an interpolation between the ferromagnetic Ising model ($p = 1$ pattern) and the Sherrington-Kirkpatrick spin glass model (many random patterns) (Kirkpatrick & Sherrington, 1978; Barra & Guerra, 2008). By encoding patterns as dynamical attractors which are robust to perturbations, HNs provide an elegant solution to pattern recognition and classification tasks. They are considered the prototypical attractor neural network, and are the historical precursor to modern recurrent neural networks.

Concurrently, spin glasses have been used extensively in the historical machine learning literature where they comprise a sub-class of "Boltzmann machines" (BMs) (Ackley et al., 1985). Given a collection of data samples drawn from a data distribution, one is generally interested in "training" a BM by tuning its weights $J_{ij}$ such that its equilibrium distribution can reproduce the data distribution as closely as possible (Hinton, 2012). The resulting optimization problem is dramatically simplified when the network has a two-layer structure where each layer has no self-interactions, so that there are only inter-layer connections (Hinton, 2012) (see Fig. 1). This architecture is known as a Restricted Boltzmann Machine (RBM), and the two layers are sometimes called the visible layer and the hidden layer. The visible layer characteristics (dimension, type of units) are determined by the training data, whereas the hidden layer can have binary or continuous units and the dimension is chosen somewhat arbitrarily. In addition to generative modelling, RBMs and their multi-layer extensions have been used for a variety of learning tasks, such as classification, feature extraction, and dimension reduction (e.g. Salakhutdinov et al. (2007); Hinton & Salakhutdinov (2006)).

There has been extensive interest in the relationship between HNs and RBMs, as both are built on the Ising model formalism and fulfill similar roles, with the aim of better understanding RBM behaviour and potentially improving performance. Various results in this area have been recently reviewed (Marullo & Agliari, 2021). In particular, an exact mapping between HNs and RBMs has been previously noted for the special case of uncorrelated (orthogonal) patterns (Barra et al., 2012). Several related models have since been studied (Agliari et al., 2013; Mézard, 2017), which partially relax the uncorrelated pattern constraint. However, the patterns observed in most real datasets exhibit significant correlations, precluding the use of these approaches.

In this paper, we demonstrate exact correspondence between HNs and RBMs in the case of correlated pattern HNs. Specifically, we show that any HN with $N$ binary units and $p < N$ arbitrary (i.e. non-orthogonal) binary patterns encoded via the projection rule (Kanter & Sompolinsky, 1987; Personnaz et al., 1986), can be transformed into an RBM with $N$ binary and $p$ gaussian variables. We then characterize when the reverse map from RBMs to HNs can be made. We consider a practical example using the mapping, and discuss the potential importance of this correspondence for the training and interpretability of RBMs.

## 2 RESULTS

We first introduce the classical solution to the problem of encoding $N$-dimensional binary $\{-1, +1\}$ vectors $\{\boldsymbol{\xi}^{\mu}\}_{\mu=1}^{p}$, termed "patterns", as global minima of a pairwise spin glass $H(\boldsymbol{s}) = -\frac{1}{2}\boldsymbol{s}^{T}\boldsymbol{J}\boldsymbol{s}$. This is often framed as a pattern retrieval problem, where the goal is to specify or learn $J_{ij}$ such that an energy-decreasing update rule for $H(\boldsymbol{s})$ converges to the patterns (i.e. they are stable fixed points). Consider the $N \times p$ matrix $\boldsymbol{\xi}$ with the $p$ patterns as its columns. Then the classical prescription known as the projection rule (or pseudo-inverse rule) (Kanter & Sompolinsky, 1987; Personnaz et al., 1986), $\boldsymbol{J} = \boldsymbol{\xi}(\boldsymbol{\xi}^{T}\boldsymbol{\xi})^{-1}\boldsymbol{\xi}^{T}$, guarantees that the $p$ patterns will be global minima of $H(\boldsymbol{s})$. This resulting spin model is commonly called a (projection) Hopfield network, and has the Hamiltonian

$$H(\boldsymbol{s}) = -\frac{1}{2}\boldsymbol{s}^{T}\boldsymbol{\xi}(\boldsymbol{\xi}^{T}\boldsymbol{\xi})^{-1}\boldsymbol{\xi}^{T}\boldsymbol{s}. \tag{1}$$

Note that $\boldsymbol{\xi}^{T}\boldsymbol{\xi}$ invertibility is guaranteed as long as the patterns are linearly independent (we therefore require $p \leq N$). Also note that in the special (rare) case of orthogonal patterns $\boldsymbol{\xi}^{\mu} \cdot \boldsymbol{\xi}^{\nu} = N\delta^{\mu\nu}$ (also called "uncorrelated"), studied in the previous work (Barra et al., 2012), one has $\boldsymbol{\xi}^{T}\boldsymbol{\xi} = N\boldsymbol{I}$ and so the pseudo-inverse interactions reduce to the well-known Hebbian form $\boldsymbol{J} = \frac{1}{N}\boldsymbol{\xi}\boldsymbol{\xi}^{T}$ (the properties of which are studied extensively in Amit et al. (1985)). Additional details on the projection HN Eq. (1) are provided in Appendix A. To make progress in analyzing Eq. (1), we first consider a transformation of $\boldsymbol{\xi}$ which eliminates the inverse factor.

### 2.1 MAPPING A HOPFIELD NETWORK TO A RESTRICTED BOLTZMANN MACHINE

In order to obtain a more useful representation of the quadratic form Eq. (1) (for our purposes), we utilize the QR-decomposition (Schott & Stewart, 1999) of $\boldsymbol{\xi}$ to "orthogonalize" the patterns,

$$\boldsymbol{\xi} = \boldsymbol{QR}, \tag{2}$$

with $\boldsymbol{Q} \in \mathbb{R}^{N \times p}, \boldsymbol{R} \in \mathbb{R}^{p \times p}$. The columns of $\boldsymbol{Q}$ are the orthogonalized patterns, and form an orthonormal basis (of non-binary vectors) for the $p$-dimensional subspace spanned by the binary patterns. $\boldsymbol{R}$ is upper triangular, and if its diagonals are held positive then $\boldsymbol{Q}$ and $\boldsymbol{R}$ are both unique (Schott & Stewart, 1999). Note both the order and sign of the columns of $\boldsymbol{\xi}$ are irrelevant for HN pattern recall, so there are $n = 2^{p} \cdot p!$ possible $\boldsymbol{Q}, \boldsymbol{R}$ pairs. Fixing a pattern ordering, we can use the orthogonality of $\boldsymbol{Q}$ to re-write the interaction matrix as

$$\boldsymbol{J} = \boldsymbol{\xi}(\boldsymbol{\xi}^{T}\boldsymbol{\xi})^{-1}\boldsymbol{\xi}^{T} = \boldsymbol{QR}(\boldsymbol{R}^{T}\boldsymbol{R})^{-1}\boldsymbol{R}^{T}\boldsymbol{Q}^{T} = \boldsymbol{QQ}^{T} \tag{3}$$

(the last equality follows from $(\boldsymbol{R}^{T}\boldsymbol{R})^{-1} = \boldsymbol{R}^{-1}(\boldsymbol{R}^{T})^{-1}$). Eq. (3) resembles the simple Hebbian rule but with *non-binary* orthogonal patterns. Defining $\boldsymbol{q} \equiv \boldsymbol{Q}^{T}\boldsymbol{s}$ in analogy to the classical pattern overlap parameter $\boldsymbol{m} \equiv \frac{1}{N}\boldsymbol{\xi}^{T}\boldsymbol{s}$ (Amit et al., 1985), we have

$$H(\boldsymbol{s}) = -\frac{1}{2}\boldsymbol{s}^{T}\boldsymbol{QQ}^{T}\boldsymbol{s} = -\frac{1}{2}\boldsymbol{q}(\boldsymbol{s}) \cdot \boldsymbol{q}(\boldsymbol{s}). \tag{4}$$

Using a Gaussian integral as in Amit et al. (1985); Barra et al. (2012); Mézard (2017) to transform (exactly) the partition function $Z \equiv \sum_{\{s\}} e^{-\beta H(s)}$ of Eq. (1), we get

$$
\begin{aligned}
Z &= \sum_{\{s\}} e^{\frac{1}{2}(\beta q)^T (\beta^{-1} I)(\beta q)} \\
&= \sum_{\{s\}} \int e^{-\frac{\beta}{2} \sum_\mu \lambda_\mu^2 + \beta \sum_\mu \lambda_\mu \sum_i Q_{i\mu} s_i} \prod_\mu \frac{d\lambda_\mu}{\sqrt{2\pi/\beta}}.
\end{aligned}
\tag{5}
$$

The second line can be seen as the partition function of an expanded Hamiltonian for the $N$ (binary) original variables $\{s_i\}$ and the $p$ (continuous) auxiliary variables $\{\lambda_\mu\}$, i.e.

$$
H_{\text{RBM}}(\{s_i\}, \{\lambda_\mu\}) = \frac{1}{2} \sum_\mu \lambda_\mu^2 - \sum_\mu \sum_i Q_{i\mu} s_i \lambda_\mu.
\tag{6}
$$

Note that this is the Hamiltonian of a binary-continuous RBM with inter-layer weights $Q_{i\mu}$. The original HN is therefore equivalent to an RBM described by Eq. (6) (depicted in Fig. 1). As mentioned above, there are many RBMs which correspond to the same HN due to the combinatorics of choosing $Q$. In fact, instead of QR factorization one can use any decomposition which satisfies $J = UU^T$, with orthogonal $U \in \mathbb{R}^{N \times p}$ (see Appendix B), in which case $U$ acts as the RBM weights. Also note the inclusion of an applied field term $-\sum_i b_i s_i$ in Eq. (1) trivially carries through the procedure, i.e. $\tilde{H}_{RBM}(\{s_i\}, \{\lambda_\mu\}) = \frac{1}{2} \sum_\mu \lambda_\mu^2 - \sum_i b_i s_i - \sum_\mu \sum_i Q_{i\mu} s_i \lambda_\mu$.

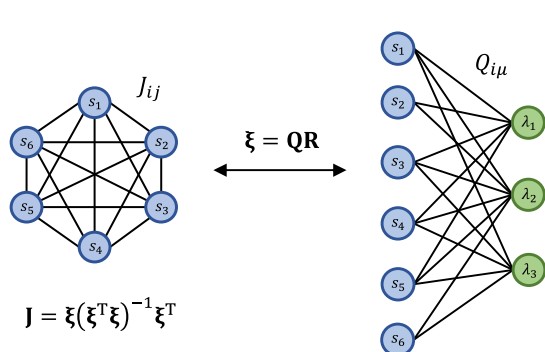

Figure 1: Correspondence between Hopfield Networks (HNs) with correlated patterns and binary-gaussian restricted boltzmann machines (RBMs). The HN has $N$ binary units and pairwise interactions $J$ defined by $p < N$ (possibly correlated) patterns $\{\xi^\mu\}_{\mu=1}^p$. The patterns are encoded as minima of Eq. (1) through the projection rule $J = \xi(\xi^T \xi)^{-1} \xi^T$, where $\xi^\mu$ form the columns of $\xi$. We orthogonalize the patterns through a QR decomposition $\xi = QR$. The HN is equivalent to an RBM with $N$ binary visible units and $p$ gaussian hidden units with inter-layer weights defined as the orthogonalized patterns $Q_{i\mu}$, and Hamiltonian Eq. (6). See (Agliari et al., 2017) for the analogous mapping in the uncorrelated case.

Instead of working with the joint form Eq. (6), one could take a different direction from Eq. (5) and sum out the original variables $\{s_i\}$, i.e.

$$
Z = \int e^{-\frac{\beta}{2} \sum_\mu \lambda_\mu^2} 2^N \prod_i \cosh\left(\beta \sum_\mu Q_{i\mu} \lambda_\mu\right) \prod_\mu \frac{d\lambda_\mu}{\sqrt{2\pi/\beta}}.
\tag{7}
$$

This continuous, $p$-dimensional representation is useful for numerical estimation of $Z$ (Section 3.1). We may write Eq. (7) as $Z = \int e^{-F_0(\lambda)} d\lambda_\mu$, where

$$
F_0(\{\lambda_\mu\}) = \frac{1}{2} \sum_\mu \lambda_\mu^2 - \frac{1}{\beta} \sum_i \ln \cosh\left(\beta \sum_\mu Q_{i\mu} \lambda_\mu\right).
\tag{8}
$$

Eq. (8) is an approximate Lyapunov function for the mean dynamics of $\{\lambda_\mu\}$; $\nabla_\lambda F_0$ describes the effective behaviour of the stochastic dynamics of the $N$ binary variables $\{s_i\}$ at temperature $\beta^{-1}$.

## 2.2 COMMENTS ON THE REVERSE MAPPING

With the mapping from HNs (with correlated patterns) to RBMs established, we now consider the reverse direction. Consider a binary-continuous RBM with inter-layer weights $W_{i\mu}$ which couple a

visible layer of $N$ binary variables $\{s_i\}$ to a hidden layer of $p$ continuous variables $\{\lambda_\mu\}$,

$$H(\boldsymbol{s}, \boldsymbol{\lambda}) = \frac{1}{2}\sum_\mu \lambda_\mu^2 - \sum_i b_i s_i - \sum_\mu \sum_i W_{i\mu} s_i \lambda_\mu. \tag{9}$$

Here we use $\boldsymbol{W}$ instead of $\boldsymbol{Q}$ for the RBM weights to emphasize that the RBM is not necessarily an HN. First, following Mehta et al. (2019), we transform the RBM to a BM with binary states by integrating out the hidden variables. The corresponding Hamiltonian for the visible units alone is (see Appendix D.1 for details),

$$\tilde{H}(\boldsymbol{s}) = -\sum_i b_i s_i - \frac{1}{2}\sum_i \sum_j \sum_\mu W_{i\mu} W_{j\mu} s_i s_j, \tag{10}$$

a pairwise Ising model with a particular coupling structure $J_{ij} = \sum_\mu W_{i\mu} W_{j\mu}$, which in vector form is

$$\boldsymbol{J} = \sum_\mu \boldsymbol{w}_\mu \boldsymbol{w}_\mu^T = \boldsymbol{W}\boldsymbol{W}^T, \tag{11}$$

where $\{\boldsymbol{w}_\mu\}$ are the $p$ columns of $\boldsymbol{W}$.

In general, this Ising model Eq. (10) produced by integrating out the hidden variables need not have Hopfield structure (discussed below). However, it automatically does (as noted in Barra et al. (2012)), in the very special case where $W_{i\mu} \in \{-1, +1\}$. In that case, the binary patterns are simply $\{\boldsymbol{w}_\mu\}$, so that Eq. (11) represents a Hopfield network with the Hebbian prescription. This situation is likely rare and may only arise as a by-product of constrained training; for a generically trained RBM the weights will not be binary. It is therefore interesting to clarify when and how real-valued RBM interactions $\boldsymbol{W}$ can be associated with HNs.

**Approximate binary representation of $\boldsymbol{W}$:** In Section 2.1, we orthogonalized the binary matrix $\boldsymbol{\xi}$ via the QR decomposition $\boldsymbol{\xi} = \boldsymbol{Q}\boldsymbol{R}$, where $\boldsymbol{Q}$ is an orthogonal (but non-binary) matrix, which allowed us to map a projection HN (defined by its patterns $\boldsymbol{\xi}$, Eq. (1)) to an RBM (defined by its inter-layer weights $\boldsymbol{Q}$, Eq. (6)).

Here we consider the reverse map. Given a trained RBM with weights $\boldsymbol{W} \in \mathbb{R}^{N \times p}$, we look for an invertible transformation $\boldsymbol{X} \in \mathbb{R}^{p \times p}$ which binarizes $\boldsymbol{W}$. We make the mild assumption that $\boldsymbol{W}$ is rank $p$. If we find such an $\boldsymbol{X}$, then $\boldsymbol{B} = \boldsymbol{W}\boldsymbol{X}$ will be the Hopfield pattern matrix (analogous to $\boldsymbol{\xi}$), with $B_{i\mu} \in \{-1, +1\}$.

This is a non-trivial problem, and an exact solution is not guaranteed. As a first step to study the problem, we relax it to that of finding a matrix $\boldsymbol{X} \in \mathrm{GL}_p(\mathbb{R})$ (i.e. invertible, $p \times p$, real) which minimizes the binarization error

$$\underset{\boldsymbol{X} \in \mathrm{GL}_p(\mathbb{R})}{\arg\min} \|\boldsymbol{W}\boldsymbol{X} - \mathrm{sgn}(\boldsymbol{W}\boldsymbol{X})\|_F. \tag{12}$$

We denote the approximately binary transformation of $\boldsymbol{W}$ via a particular solution $\boldsymbol{X}$ by

$$\boldsymbol{B}_p = \boldsymbol{W}\boldsymbol{X}. \tag{13}$$

We also define the associated error matrix $\boldsymbol{E} \equiv \boldsymbol{B}_p - \mathrm{sgn}(\boldsymbol{B}_p)$. We stress that $\boldsymbol{B}_p$ is non-binary and approximates $\boldsymbol{B} \equiv \mathrm{sgn}(\boldsymbol{B}_p)$, the columns of which will be HN patterns under certain conditions on $\boldsymbol{E}$. We provide an initial characterization and example in Appendix D.

## 3 EXPERIMENTS ON MNIST DATASET

Next we investigate whether the Hopfield-RBM correspondence can provide an advantage for training binary-gaussian RBMs. We consider the popular MNIST dataset of handwritten digits (LeCun et al., 1998) which consists of $28 \times 28$ images of handwritten images, with greyscale pixel values $0$ to $255$. We treat the sample images as $N \equiv 784$ dimensional binary vectors of $\{-1, +1\}$ by setting all non-zero values to $+1$. The dataset includes $M \equiv 60,000$ training images and $10,000$ testing images, as well as their class labels $\mu \in \{0, \ldots, 9\}$.

### 3.1 GENERATIVE OBJECTIVE

The primary task for generative models such as RBMs is to reproduce a data distribution. Given a data distribution $p_{\text{data}}$, the *generative objective* is to train a model (here an RBM defined by its parameters $\boldsymbol{\theta}$), such that the model distribution $p_{\boldsymbol{\theta}}$ is as close to $p_{\text{data}}$ as possible. This is often quantified by the Kullback-Leibler (KL) divergence $D_{KL}(p_{\text{data}}\|p_{\boldsymbol{\theta}}) = \sum_{\boldsymbol{s}} p_{\text{data}}(\boldsymbol{s}) \ln\left(\frac{p_{\text{data}}(\boldsymbol{s})}{p_{\boldsymbol{\theta}}(\boldsymbol{s})}\right)$. One generally does not have access to the actual data distribution, instead there is usually a representative training set $S = \{\boldsymbol{s}_a\}_{a=1}^{M}$ sampled from it. As the data distribution is constant with respect to $\boldsymbol{\theta}$, the generative objective is equivalent to maximizing $L(\boldsymbol{\theta}) = \frac{1}{M}\sum_a \ln p_{\boldsymbol{\theta}}(\boldsymbol{s}_a)$.

#### 3.1.1 HOPFIELD RBM SPECIFICATION

With labelled classes of training data, specification of an RBM via a one-shot Hopfield rule ("Hopfield RBM") is straightforward. In the simplest approach, we define $p = 10$ representative patterns via the (binarized) class means

$$\boldsymbol{\xi}^{\mu} \equiv \text{sgn}\left(\frac{1}{|S_{\mu}|}\sum_{\boldsymbol{s}\in S_{\mu}}\boldsymbol{s}\right). \tag{14}$$

where $\mu \in \{0, \ldots, 9\}$ and $S_{\mu}$ is the set of sample images for class $\mu$. These patterns comprise the columns of the $N \times p$ pattern matrix $\boldsymbol{\xi}$, which is then orthogonalized as in Eq. (2) to obtain the RBM weights $\boldsymbol{W}$ which couple $N$ binary visible units to $p$ gaussian hidden units.

We also consider refining this approach by considering sub-classes within each class, representing, for example, the different ways one might draw a "7". As a proof of principle, we split each digit class into $k$ sub-patterns using hierarchical clustering. We found good results with Agglomerative clustering using Ward linkage and Euclidean distance (see Murtagh & Contreras (2012) for an overview of this and related methods). In this way, we can define a hierarchy of Hopfield RBMs. At one end, $k = 1$, we have our simplest RBM which has $p = 10$ hidden units and encodes 10 patterns (using Eq. (14)), one for each digit class. At the other end, $10k/N \to 1$, we can specify increasingly refined RBMs that encode $k$ sub-patterns for each of the 10 digit classes, for a total of $p = 10k$ patterns and hidden units. This approach has an additional cost of identifying the sub-classes, but is still typically faster than training the RBM weights directly (discussed below).

The generative performance as a function of $k$ and $\beta$ is shown in Fig. 2, and increases monotonically with $k$ in the range plotted. If $\beta$ is too high (very low temperature) the free energy basins will be very deep directly at the patterns, and so the model distribution will not capture the diversity of images from the data. If $\beta$ is too low (high temperature), there is a "melting transition" where the original pattern basins disappear entirely, and the data will therefore be poorly modelled. Taking $\alpha = p/N \sim 0.1$ (roughly $k = 8$), Fig. 1 of Kanter & Sompolinsky (1987) predicts $\beta_m \approx 1.5$ for the theoretical melting transition for the pattern basins. Interestingly, this is quite close to our observed peak near $\beta = 2$. Note also as $k$ is increased, the generative performance is sustained at lower temperatures.

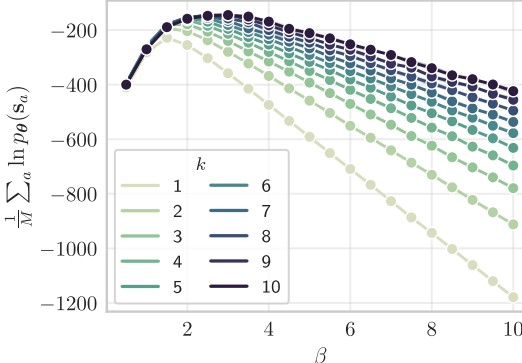

Figure 2: Hopfield RBM generative performance as a function of $\beta$ for varying numbers of encoded sub-patterns $k$ per digit. The number of hidden units in each RBM is $p = 10k$, corresponding to the total number of encoded patterns. $\ln Z$ is computed using annealed importance sampling (AIS) (Neal, 2001) on the continuous representation of $Z$, Eq. (7), with 500 chains for 1000 steps (see Appendix E). Each curve displays the mean of three runs.

In situations where one already has access to the class labels, this approach to obtain RBM weights is very fast. The class averaging has negligible computational cost $O(MN)$ for the whole training

set ($M$ samples), and the QR decomposition has a modest complexity of $O(Np^2)$ (Schott & Stewart, 1999). Conventional RBM training, discussed below, requires significantly more computation.

### 3.1.2 CONVENTIONAL RBM TRAINING

RBM training is performed through gradient ascent on the log-likelihood of the data, $L(\boldsymbol{\theta}) = \frac{1}{M}\sum_a \ln p_{\boldsymbol{\theta}}(\boldsymbol{s}_a)$ (equivalent here to minimizing KL divergence, as mentioned above). We are focused here on the weights $\boldsymbol{W}$ in order to compare to the Hopfield RBM weights, and so we neglect the biases on both layers. As is common (Hinton, 2012), we approximate the total gradient by splitting the training dataset into "mini-batches", denoted $B$. The resulting gradient ascent rule for the weights is (see Appendix E)

$$W_{i\mu}^{t+1} = W_{i\mu}^t + \eta\left(\left\langle\sum_j W_{j\mu}s_i^a s_j^a\right\rangle_{a\in B} - \langle s_i\lambda_\mu\rangle_{\text{model}}\right), \tag{15}$$

where $\langle s_i\lambda_\mu\rangle_{\text{model}} \equiv Z^{-1}\sum_{\{\boldsymbol{s}\}}\int s_i\lambda_\mu e^{-\beta H(\boldsymbol{s},\boldsymbol{\lambda})}\prod_\mu d\lambda_\mu$ is an average over the model distribution.

The first bracketed term of Eq. (15) is simple to calculate at each iteration of the weights. The second term, however, is intractable as it requires one to calculate the partition function $Z$. We instead approximate it using contrastive divergence (CD-$K$) (Carreira-Perpinan & Hinton, 2005; Hinton, 2012). See Appendix E for details. Each full step of RBM weight updates involves $O(KBNp)$ operations (Melchior et al., 2017). Training generally involves many mini-batch iterations, such that the entire dataset is iterated over (one epoch) many times. In our experiments we train for 50 epochs with mini-batches of size 100 ($3\cdot10^5$ weight updates), so the overall training time can be extensive compared to the one-shot Hopfield approach presented above. For further details on RBM training see e.g. Hinton (2012); Melchior et al. (2017).

In Fig. 3, we give an example of the Hopfield RBM weights (for $k = 1$), as well as how they evolve during conventional RBM training. Note Fig. 3(a), (b) appear qualitatively similar, suggesting that the proposed initialization $\boldsymbol{Q}$ from Eqs. (2), (14) may be near a local optimum of the objective.

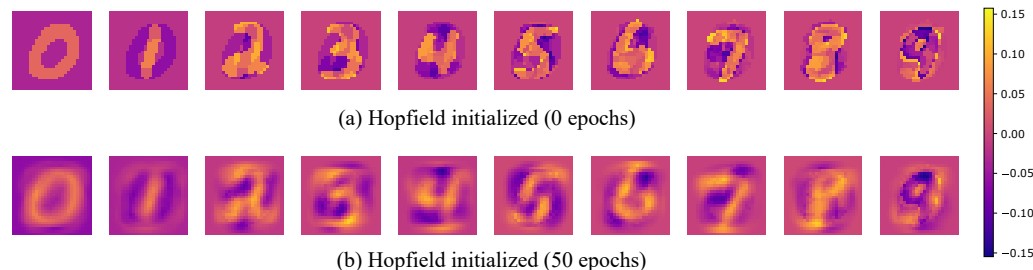

(a) Hopfield initialized (0 epochs)

(b) Hopfield initialized (50 epochs)

Figure 3: Binary-gaussian RBM weights for $p = 10$ hidden units prior to and during generative training. (a) Initial values of the columns of $\boldsymbol{W}$ (specified as the orthogonalized Hopfield patterns via Eqs. (2), (14)). (b) Same columns of $\boldsymbol{W}$ after 50 epochs of CD-20 training (see Fig. 4(a)).

In Fig. 4(a), (b), we compare conventional RBM training on four different weight initializations: (i) random $W_{i\mu} \sim \mathcal{N}(0, 0.01)$ (purple), commonly used in the literature; (ii) our proposed weights from the projection rule Hopfield mapping for correlated patterns (blue); (iii) the "Hebbian" Hopfield mapping described in previous work for uncorrelated patterns, $\boldsymbol{W} = N^{-1/2}\boldsymbol{\xi}$ (Barra et al., 2012) (green); and (iv) the top $p$ PCA components of the training data (pink). In Fig. 4(c), (d) we compare generated sample images from two RBMs, each with $p = 50$ hidden units but different initial weights (random in (c) and the HN mapping in (d)). The quality of samples in Fig. 4(d) reflect the efficient training of the Hopfield initialized RBM.

Fig. 4(a), (b) show that the Hopfield initialized weights provide an advantage over other approaches during the early stages of training. The PCA and Hebbian initializations start at much lower values of the objective and require one or more epochs of training to perform similarly (Fig. 4(a) inset), while the randomly initialized RBM takes $> 25$ epochs to reach a similar level. All initializations ultimately reach the same value. This is noteworthy because the proposed weight initialization is fast compared to conventional RBM training. PCA performs best for intermediate training times.

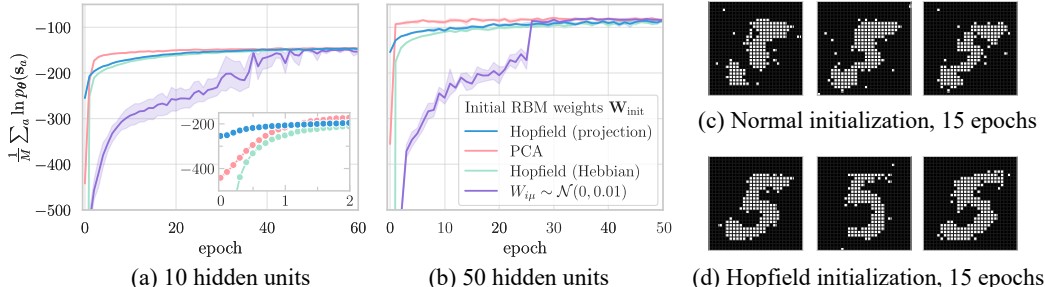

(c) Normal initialization, 15 epochs

(a) 10 hidden units     (b) 50 hidden units     (d) Hopfield initialization, 15 epochs

Figure 4: Generative performance of binary-gaussian RBMs trained with (a) $p = 10$ and (b) $p = 50$ hidden units. Curves are colored according to the choice of weight initialization (see legend in (b), and further detail in the preceding text). Each curve shows the mean and standard deviation over 5 runs. The inset in (a) details the first two epochs. We compute $\ln p_{\boldsymbol{\theta}}$ as in Fig. 2, but with 100 AIS chains. The learning rate is $\eta_0 = 10^{-4}$ except the first 25 epochs of the randomly initialized weights in (b), where we used $\eta = 5\eta_0$ due to slow training. The mini-batch size is $B = 100$ for all curves in (b) and the purple curve in (a), and $B = 1000$ otherwise. (c), (d) Samples from two RBMs from (b) (projection HN and random) after 15 epochs, generated by initializing the visible state to an example image from the desired class and performing 20 RBM updates with $\beta = 2$. Training parameters: $\beta = 2$, and CD-20.

Despite being a common choice, the random initialization trains surprisingly slowly, taking roughly 40 epochs in Fig. 4(a), and in Fig. 4(b) we had to increase the basal learning rate $\eta_0 = 10^{-4}$ by a factor of 5 for the first 25 epochs due to slow training. The non-random initializations, by comparison, arrive at the same maximum value much sooner. The relatively small change over training for the Hopfield initialized weights supports the idea that they may be near a local optimum of the objective, and that conventional training may simply be mildly tuning them (Fig. 3).

That the HN initialization performs well at 0 epochs suggests that the $p$ Hopfield patterns concisely summarize the dataset. This is intuitive, as the projection rule encodes the patterns (and nearby states) as high probability basins in the free energy landscape of Eq. (1). As the data itself is clustered near the patterns, these basins should model the true data distribution well. Overall, our results suggest that the HN correspondence provides a useful initialization for generative modelling with binary-gaussian RBMs, displaying excellent performance with minimal training.

## 3.2 CLASSIFICATION OBJECTIVE

As with the generative objective, we find that the Hopfield initialization provides an advantage for classification tasks. Here we consider the closely related MNIST classification problem. The goal is to train a model on the MNIST Training dataset which accurately predicts the class of presented images. The key statistic is the number of misclassified images on the MNIST Testing dataset.

We found relatively poor classification results with the single (large) RBM architecture from the preceding Section 3.1. Instead, we use a minimal product-of-experts (PoE) architecture as described in Hinton (2002): the input data is first passed to 10 RBMs, one for each class $\mu$. This "layer of RBMs" functions as a pre-processing layer which maps the high dimensional sample $\boldsymbol{s}$ to a feature vector $\boldsymbol{f}(\boldsymbol{s}) \in \mathbb{R}^{10}$. This feature vector is then passed to a logistic regression layer in order to predict the class of $\boldsymbol{s}$. The RBM layer and the classification layer are trained separately.

The first step is to train the RBM layer to produce useful features for classification. As in Hinton (2002), each small RBM is trained to model the distribution of samples from a specific digit class $\mu$. We use CD-20 generative training as in Section 3.1, with the caveat that each expert is trained solely on examples from their respective class. Each RBM connects $N$ binary visible units to $k$ gaussian hidden units, and becomes an "expert" at generating samples from one class. To focus on the effect of interlayer weight initialization, we set the layer biases to 0.

After generative training, each expert should have relatively high probability $p_{\boldsymbol{\theta}}^{(\mu)}(\boldsymbol{s}_a)$ for sample digits $\boldsymbol{s}_a$ of the corresponding class $\mu$, and lower probability for digits from other classes. This idea is used to define 10 features, one from each expert, based on the log-probability of a given sample

under each expert, $\ln p_{\boldsymbol{\theta}}^{(\mu)}(\boldsymbol{s}_a) = -\beta H^{(\mu)}(\boldsymbol{s}_a) - \ln Z^{(\mu)}$. Note that $\beta$ and $\ln Z^{(\mu)}$ are constants with respect to the data and thus irrelevant for classification. For a binary-gaussian RBM, $H^{(\mu)}(\boldsymbol{s}_a)$ has the simple form Eq. (10), so the features we use are

$$f^{(\mu)}(\boldsymbol{s}) = \sum_i \sum_j \sum_\nu W_{i\nu}^{(\mu)} W_{j\nu}^{(\mu)} s_i s_j = ||\boldsymbol{s}^T \boldsymbol{W}^{(\mu)}||^2. \qquad (16)$$

With the feature map defined, we then train a standard logistic regression classifier (using scikit-learn (Pedregosa et al., 2011)) on these features. In Fig. 5, we report the classification error on the MNIST Testing set of 10,000 images (held out during both generative and classification training). Note the size $p = 10$ of the feature vector is independent of the hidden dimension $k$ of each RBM, so the classifier is very efficient.

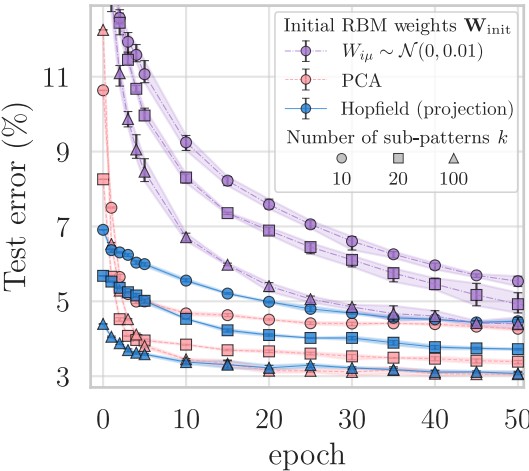

Figure 5: Product-of-experts classification performance for the various weight initializations. For each digit model (expert), we perform CD-20 training according to Eq. (15) (as in Fig. 4) for a fixed number of epochs. A given sample image is mapped to the 10-dimensional feature vector with elements Eq. (16). These features are used to train a logistic regression classifier, and the average MNIST Test set errors are reported. The initializations considered for each expert's weights $\boldsymbol{W}_{\text{init}}^{(\mu)} \in \mathbb{R}^{N \times k}$ are $W_{i\mu} \sim \mathcal{N}(0, 0.01)$ (purple, dash-dot), PCA (pink, dashed), and the orthogonalized Hopfield sub-patterns for digit class $\mu$ (blue, solid). The error bars show the min/max of three runs.

Despite this relatively simple approach, the PoE initialized using the orthogonalized Hopfield patterns ("Hopfield PoE") performs fairly well (Fig. 5, blue curves), especially as the number of sub-patterns is increased. We found that generative training beyond 50 epochs did not significantly improve performance for the projection HN or PCA. (in Fig. E.1, we train to 100 epochs and also display the aforementioned "Hebbian" initial condition, which performs much worse for classification). Intuitively, increasing the number of hidden units $k$ increases classification performance independent of weight initialization (with sufficient training).

For $k$ fixed, the Hopfield initialization provides a significant benefit to classification performance compared to the randomly initialized weights (purple curves). For few sub-patterns (circles $k = 10$ and squares $k = 20$), the Hopfield initialized models perform best without additional training and until 1 epoch, after which PCA (pink) performs better. When each RBM has $k = 100$ hidden features (triangles), the Hopfield and PCA PoE reach $3.0\%$ training error, whereas the randomly initialized PoE reaches $4.5\%$. However, the Hopfield PoE performs much better than PCA with minimal training, and maintains its advantage until 10 epochs, after which they are similar. Interestingly, both the Hopfield and PCA initialized PoE with just $k = 10$ encoded patterns performs better than or equal to the $k = 100$ randomly initialized PoE at each epoch despite having an order of magnitude fewer trainable parameters. Finally, without any generative training (0 epochs), the $k = 100$ Hopfield PoE performs slightly better ($4.4\%$) than the $k = 100$ randomly initialized PoE with 50 epochs of training.

## 4 Discussion

We have presented an explicit, exact mapping from projection rule Hopfield networks to Restricted Boltzmann Machines with binary visible units and gaussian hidden units. This provides a generalization of previous results which considered uncorrelated patterns (Barra et al., 2012), or special cases of correlated patterns (Agliari et al., 2013; Mézard, 2017). We provide an initial characterization of the reverse map from RBMs to HNs, along with a matrix-factorization approach to construct approximate associated HNs when the exact reverse map is not possible. Importantly, our HN to

RBM mapping can be applied to correlated patterns such as those found in real world datasets. As a result, we are able to conduct experiments (Section 3) on the MNIST dataset which suggest the mapping provides several advantages.

The conversion of an HN to an equivalent RBM has practical utility: it trades simplicity of presentation for faster processing. The weight matrix of the RBM is potentially much smaller than the HN ($Np$ elements instead of $N(N-1)/2$). More importantly, proper sampling of stochastic trajectories in HNs requires asynchronous updates of the units, whereas RBM dynamics can be simulated in a parallelizable, layer-wise fashion. We also utilized the mapping to efficiently estimate the partition function of the Hopfield network (Fig. 2) by summing out the spins after representing it as an RBM.

This mapping also has another practical utility. When used as an RBM weight initialization, the HN correspondence enables efficient training of generative models (Section 3.1, Fig. 4). RBMs initialized with random weights and trained for a moderate amount of time perform worse than RBMs initialized to orthogonalized Hopfield patterns and not trained at all. Further, with mild training of just a few epochs, Hopfield RBMs outperform conventionally initialized RBMs trained several times longer. The revealed initialization also shows advantages over alternative non-random initializations (PCA and the "Hebbian" Hopfield mapping) during early training. By leveraging this advantage for generative tasks, we show that the correspondence can also be used to improve classification performance (Section 3.2, Fig. 5, Appendix E.3).

Overall, the RBM initialization revealed by the mapping allows for smaller models which perform better despite shorter training time (for instance, using fewer hidden units to achieve similar classification performance). Reducing the size and training time of models is critical, as more realistic datasets (e.g. gene expression data from single-cell RNA sequencing) may require orders of magnitude more visible units. For generative modelling of such high dimensional data, our proposed weight initialization based on orthogonalized Hopfield patterns could be of practical use. Our theory and experiments are a proof-of-principle; if they can be extended to the large family of deep architectures which are built upon RBMs, such as deep belief networks (Hinton et al., 2006) and deep Boltzmann machines (Salakhutdinov & Hinton, 2009), it would be of great benefit. This will be explored in future work.

More broadly, exposing the relationship between RBMs and their representative HNs helps to address the infamous interpretability problem of machine learning which criticizes trained models as "black boxes". HNs are relatively transparent models, where the role of the patterns as robust dynamical attractors is theoretically well-understood. We believe this correspondence, along with future work to further characterize the reverse map, will be especially fruitful for explaining the performance of deep architectures constructed from RBMs.

## ACKNOWLEDGMENTS

The authors thank Duncan Kirby and Jeremy Rothschild for helpful comments and discussions. This work is supported by the National Science and Engineering Research Council of Canada (NSERC) through Discovery Grant RGPIN 402591 to A.Z. and CGS-D Graduate Fellowship to M.S.

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

## A   Hopfield network details

Consider $p < N$ $N$-dimensional binary patterns $\{\boldsymbol{\xi}^\mu\}_{\mu=1}^p$ that are to be "stored". From them, construct the $N \times p$ matrix $\boldsymbol{\xi}$ whose columns are the $p$ patterns. If they are mutually orthogonal (e.g. randomly sampled patterns in the large $N \to \infty$ limit), then choosing interactions according to the Hebbian rule, $\boldsymbol{J}_{\text{Hebb}} = \frac{1}{N}\boldsymbol{\xi}\boldsymbol{\xi}^T$, guarantees that they will all be stable minima of $H(\boldsymbol{s}) = -\frac{1}{2}\boldsymbol{s}^T \boldsymbol{J}\boldsymbol{s}$, provided $\alpha \equiv p/N < \alpha_c$, where $\alpha_c \approx 0.14$ (Amit et al., 1985). If they are not mutually orthogonal (referred to as *correlated*), then using the "projection rule" $\boldsymbol{J}_{\text{Proj}} = \boldsymbol{\xi}(\boldsymbol{\xi}^T\boldsymbol{\xi})^{-1}\boldsymbol{\xi}^T$ guarantees that they will all be stable minima of $H(\boldsymbol{s})$, provided $p < N$ (Kanter & Sompolinsky, 1987; Personnaz et al., 1986). Note $\boldsymbol{J}_{\text{Proj}} \to \boldsymbol{J}_{\text{Hebb}}$ in the limit of orthogonal patterns. In the main text, we use $\boldsymbol{J}$ as shorthand for $\boldsymbol{J}_{\text{Proj}}$.

We provide some relevant notation from Kanter & Sompolinsky (1987). Define the *overlap* of a state $\boldsymbol{s}$ with the $p$ patterns as $\boldsymbol{m}(\boldsymbol{s}) \equiv \frac{1}{N}\boldsymbol{\xi}^T\boldsymbol{s}$, and define the *projection* of a state $\boldsymbol{s}$ onto the $p$ patterns as $\boldsymbol{a}(\boldsymbol{s}) \equiv (\boldsymbol{\xi}^T\boldsymbol{\xi})^{-1}\boldsymbol{\xi}^T\boldsymbol{s} \equiv \boldsymbol{A}^{-1}\boldsymbol{m}(\boldsymbol{s})$. Note $\boldsymbol{A} \equiv \boldsymbol{\xi}^T\boldsymbol{\xi}$ is the overlap matrix, and $\boldsymbol{m}_\mu, \boldsymbol{a}_\mu \in [-1, 1]$.

We can re-write the projection rule Hamiltonian Eq. (1) as

$$H(\boldsymbol{s}) = -\frac{N}{2}\boldsymbol{m}(\boldsymbol{s}) \cdot \boldsymbol{a}(\boldsymbol{s}). \tag{A.1}$$

For simplicity, we include the self-interactions rather than keeping track of their omission; the results are the same as $N \to \infty$. From Eq. (A.1), several quadratic forms can be written depending on which variables one wants to work with:

I. $H(\boldsymbol{s}) = -\frac{N^2}{2}\boldsymbol{m}^T(\boldsymbol{\xi}^T\boldsymbol{\xi})^{-1}\boldsymbol{m}$

II. $H(\boldsymbol{s}) = -\frac{1}{2}\boldsymbol{a}^T(\boldsymbol{\xi}^T\boldsymbol{\xi})\boldsymbol{a}$

These are the starting points for the alternative Boltzmann Machines (i.e. not RBMs) presented in Appendix C.

## B    ADDITIONAL HN TO RBM MAPPINGS

We used QR factorization in the main text to establish the HN to RBM mapping. However, one can use any decomposition which satisfies

$$\boldsymbol{J}_{\text{Proj}} = \boldsymbol{U}\boldsymbol{U}^T \tag{B.1}$$

such that $\boldsymbol{U} \in \mathbb{R}^{N \times p}$ is orthogonal (*orthogonal* for tall matrices means $\boldsymbol{U}^T\boldsymbol{U} = \boldsymbol{I}$). In that case, $\boldsymbol{U}$ becomes the RBM weights. We provide two simple alternatives below, and show they are all part of the same family of orthogonal decompositions.

*"Square root" decomposition*: Define the matrix $\boldsymbol{K} \equiv \boldsymbol{\xi}(\boldsymbol{\xi}^T\boldsymbol{\xi})^{-1/2}$. Note that $\boldsymbol{K}$ is orthogonal, and that $\boldsymbol{J}_{\text{Proj}} = \boldsymbol{K}\boldsymbol{K}^T$.

*Singular value decomposition:* More generally, consider the SVD of the pattern matrix $\boldsymbol{\xi}$:

$$\boldsymbol{\xi} = \boldsymbol{U}\boldsymbol{\Sigma}\boldsymbol{V}^T \tag{B.2}$$

where $\boldsymbol{U} \in \mathbb{R}^{N \times p}$, $\boldsymbol{V} \in \mathbb{R}^{p \times p}$ store the left and right singular vectors (respectively) of $\boldsymbol{\xi}$ as orthogonal columns, and $\boldsymbol{\Sigma} \in \mathbb{R}^{p \times p}$ is diagonal and contains the singular values of $\boldsymbol{\xi}$. Note in the limit of orthogonal patterns, we have $\boldsymbol{\Sigma} = \sqrt{N}\boldsymbol{I}$. This decomposition gives several relations for quantities of interest:

$$\begin{aligned} \boldsymbol{A} &\equiv \boldsymbol{\xi}^T\boldsymbol{\xi} & &= \boldsymbol{V}\boldsymbol{\Sigma}^2\boldsymbol{V}^T \\ \boldsymbol{J}_{\text{Hebb}} &\equiv \frac{1}{N}\boldsymbol{\xi}\boldsymbol{\xi}^T & &= \frac{1}{N}\boldsymbol{U}\boldsymbol{\Sigma}^2\boldsymbol{U}^T \\ \boldsymbol{J}_{\text{Proj}} &\equiv \boldsymbol{\xi}(\boldsymbol{\xi}^T\boldsymbol{\xi})^{-1}\boldsymbol{\xi}^T & &= \boldsymbol{U}\boldsymbol{U}^T. \end{aligned} \tag{B.3}$$

The last line is simply the diagonalization of $\boldsymbol{J}_{\text{Proj}}$, and shows that our RBM mapping is preserved if we swap the $\boldsymbol{Q}$ from QR with $\boldsymbol{U}$ from SVD. However, since there are $p$ degenerate eigenvalues $\sigma^2 = 1$, $\boldsymbol{U}$ is not unique - any orthogonal basis for the 1-eigenspace can be chosen. Thus $\boldsymbol{U}' = \boldsymbol{U}\boldsymbol{O}$ where $\boldsymbol{O}$ is orthogonal is also valid, and the QR decomposition and "square root" decomposition correspond to particular choices of $\boldsymbol{O}$.

## C    HN TO BM MAPPINGS USING ALTERNATIVE REPRESENTATIONS OF THE HOPFIELD HAMILTONIAN

In addition to the orthogonalized representation in the main text, there are two natural representations to consider based on the pattern overlaps and projections introduced in Appendix A. These lead to generalized Boltzmann Machines (BMs) consisting of $N$ original binary spins and $p$ continuous variables. These representations are not RBMs as the continuous variables interact with each other. We present them for completeness.

*"Overlap" Boltzmann Machine*: Writing $H(\boldsymbol{s}) = -\frac{N^2}{2}\boldsymbol{m}^T(\boldsymbol{\xi}^T\boldsymbol{\xi})^{-1}\boldsymbol{m}$, we have

$$Z = \sum_{\{\boldsymbol{s}\}} \exp\left(\frac{1}{2}(\beta\sqrt{N}\boldsymbol{m})^T \left(\frac{\beta}{N}\boldsymbol{\xi}^T\boldsymbol{\xi}\right)^{-1}(\beta\sqrt{N}\boldsymbol{m})\right). \tag{C.1}$$

Applying the gaussian integral identity,

$$Z = \sqrt{\det(\boldsymbol{\xi}^T\boldsymbol{\xi})} \sum_{\{\boldsymbol{s}\}} \int e^{-\frac{\beta}{2N}\sum_{\mu,\nu}(\boldsymbol{\xi}^T\boldsymbol{\xi})_{\mu\nu}\lambda_\mu\lambda_\nu + \frac{\beta}{\sqrt{N}}\sum_\mu \lambda_\mu \sum_i \xi_{i\mu}s_i} \prod_\mu \frac{d\lambda_\mu}{\sqrt{2\pi N/\beta}}, \tag{C.2}$$

from which we identify the BM Hamiltonian

$$H(\boldsymbol{s}, \boldsymbol{\lambda}) = \frac{1}{2N}\sum_{\mu,\nu}(\boldsymbol{\xi}^T\boldsymbol{\xi})_{\mu\nu}\lambda_\mu\lambda_\nu - \frac{1}{\sqrt{N}}\sum_\mu \sum_i \xi_{i\mu}s_i\lambda_\mu. \tag{C.3}$$

This is the analog of Eq. (6) in the main text for the "overlap" representation. Note we can also sum out the binary variables in Eq. (C.2), which allows for an analogous expression to Eq. (8),

$$F_0(\{\lambda_\mu\}) = \frac{1}{2N}\sum_{\mu,\nu}(\boldsymbol{\xi}^T\boldsymbol{\xi})_{\mu\nu}\lambda_\mu\lambda_\nu - \frac{1}{\beta}\sum_i \ln\cosh\left(\frac{\beta}{\sqrt{N}}\sum_\mu \xi_{i\mu}\lambda_\mu\right). \tag{C.4}$$

Curiously, we note that we may perform a second Gaussian integral on Eq. (C.2), introducing new auxiliary variables $\tau_\nu$ to remove the interactions between the $\lambda_\mu$ variables:

$$Z = \sqrt{\det\left(\frac{\beta^2}{N}\boldsymbol{\xi}^T\boldsymbol{\xi}\right)} \sum_{\{s\}} \int \int e^{-\frac{\beta}{2}\boldsymbol{\tau}^T\boldsymbol{\tau} + \frac{\beta}{\sqrt{N}}\boldsymbol{\lambda}^T\boldsymbol{\xi}^T\boldsymbol{s} + i\frac{\beta}{\sqrt{N}}\boldsymbol{\lambda}^T(\boldsymbol{\xi}^T\boldsymbol{\xi})^{1/2}\boldsymbol{\tau}} \prod_\nu \frac{d\tau_\nu}{\sqrt{2\pi}} \prod_\mu \frac{d\lambda_\mu}{\sqrt{2\pi}}. \quad (C.5)$$

Eq. (C.5) describes a three-layer RBM with complex interactions between the $\lambda_\mu$ and $\tau_\nu$ variables, a representation which could be useful in some contexts.

*"Projection" Boltzmann Machine*: Proceeding as above but for $H(\boldsymbol{s}) = -\frac{1}{2}\boldsymbol{a}^T(\boldsymbol{\xi}^T\boldsymbol{\xi})\boldsymbol{a}$, one finds

$$Z = \det\left(\boldsymbol{\xi}^T\boldsymbol{\xi}\right)^{-1/2} \sum_{\{s\}} \int e^{-\frac{\beta}{2}\boldsymbol{\lambda}^T(\boldsymbol{\xi}^T\boldsymbol{\xi})^{-1}\boldsymbol{\lambda} + \beta\boldsymbol{\lambda}^T(\boldsymbol{\xi}^T\boldsymbol{\xi})^{-1}\boldsymbol{\xi}^T\boldsymbol{s}} \prod_\mu \frac{d\lambda_\mu}{\sqrt{2\pi/\beta}}, \quad (C.6)$$

which corresponds to the BM Hamiltonian

$$H(\boldsymbol{s}, \boldsymbol{\lambda}) = \frac{1}{2}\boldsymbol{\lambda}^T(\boldsymbol{\xi}^T\boldsymbol{\xi})^{-1}\boldsymbol{\lambda} - \boldsymbol{\lambda}^T(\boldsymbol{\xi}^T\boldsymbol{\xi})^{-1}\boldsymbol{\xi}^T\boldsymbol{s}. \quad (C.7)$$

The analogous expression to Eq. (8) in this case is

$$F_0(\boldsymbol{\lambda}) = \frac{1}{2}\boldsymbol{\lambda}^T(\boldsymbol{\xi}^T\boldsymbol{\xi})^{-1}\boldsymbol{\lambda} - \frac{1}{\beta}\sum_i \ln\cosh\left(\beta[\boldsymbol{\xi}(\boldsymbol{\xi}^T\boldsymbol{\xi})^{-1}\boldsymbol{\lambda}]_i\right). \quad (C.8)$$

# D   RBM TO HN DETAILS

## D.1   INTEGRATING OUT THE HIDDEN VARIABLES

The explanation from Mehta et al. (2019) for integrating out the hidden variables of an RBM is presented here for completeness. For a given binary-gaussian RBM defined by $H_{\text{RBM}}(\boldsymbol{s}, \boldsymbol{\lambda})$ (as in Eq. (9)), we have $p(\boldsymbol{s}) = Z^{-1}\int e^{-\beta H_{\text{RBM}}(\boldsymbol{s}, \boldsymbol{\lambda})}d\boldsymbol{\lambda}$. Consider also that $p(\boldsymbol{s}) = Z^{-1}e^{-\beta H(\boldsymbol{s})}$ for some unknown $H(\boldsymbol{s})$. Equating these expressions gives,

$$H(\boldsymbol{s}) = -\sum_i b_i s_i - \frac{1}{\beta}\sum_\mu \ln\left(\int e^{-\beta\frac{1}{2}\lambda_\mu^2 + \beta\sum_i W_{i\mu}s_i\lambda_\mu}d\lambda_\mu\right). \quad (D.1)$$

Decompose the argument of $\ln(\cdot)$ in Eq. (D.1) by defining $q_\mu(\lambda_\mu)$, a gaussian with zero mean and variance $\beta^{-1}$. Writing $t_\mu = \beta\sum_i W_{i\mu}s_i$, one observes that the second term (up to a constant) is a sum of cumulant generating functions, i.e.

$$K_\mu(t_\mu) \equiv \ln\langle e^{t_\mu\lambda_\mu}\rangle_{q_\mu} = \ln\left(\int q_\mu e^{t_\mu\lambda_\mu}d\lambda_\mu\right). \quad (D.2)$$

These can be written as a cumulant expansion, $K_\mu(t_\mu) = \sum_{n=1}\kappa_\mu^{(n)}\frac{t_\mu^n}{n!}$, where $\kappa_\mu^{(n)} = \frac{\partial K_\mu}{\partial t_\mu}|_{t_\mu=0}$ is the $n^{\text{th}}$ cumulant of $q_\mu$. However, since $q_\mu(\lambda_\mu)$ is gaussian, only the second term remains, leaving $K_\mu(t_\mu) = \frac{1}{\beta}\frac{(\beta\sum_i W_{i\mu}s_i)^2}{2}$. Putting this all together, we have

$$H(\boldsymbol{s}) = -\sum_i b_i s_i - \frac{1}{2}\sum_i\sum_j\sum_\mu W_{i\mu}W_{j\mu}s_i s_j, \quad (D.3)$$

Note that in general, $q_\mu(\lambda_\mu)$ need not be gaussian, in which case the resultant Hamiltonian $H(\boldsymbol{s})$ can have higher order interactions (expressed via the cumulant expansion).

## D.2   APPROXIMATE REVERSE MAPPING

Suppose one has a trial solution $\boldsymbol{B}_p = \boldsymbol{W}\boldsymbol{X}$ to the approximate binarization problem Eq. (12), with error matrix $\boldsymbol{E} \equiv \boldsymbol{B}_p - \text{sgn}(\boldsymbol{B}_p)$. We consider two cases depending on if $\boldsymbol{W}$ is orthogonal.

**Case 1:** If $\boldsymbol{W}$ is orthogonal, then starting from Eq. (11) and applying Eq. (13), we have $\boldsymbol{J} = \boldsymbol{W}\boldsymbol{W}^T = \boldsymbol{B}_p(\boldsymbol{X}^T\boldsymbol{X})^{-1}\boldsymbol{B}_p^T$. Using $\boldsymbol{I} = \boldsymbol{W}^T\boldsymbol{W}$, we get

$$\boldsymbol{J} = \boldsymbol{B}_p(\boldsymbol{B}_p^T\boldsymbol{B}_p)^{-1}\boldsymbol{B}_p^T. \tag{D.4}$$

Thus, the interactions between the visible units is the familiar projection rule used to store the approximately binary patterns $\boldsymbol{B}_p$. "Storage" means the patterns are stable fixed points of the deterministic update rule $\boldsymbol{s}^{t+1} \equiv \mathrm{sgn}(\boldsymbol{J}\boldsymbol{s}^t)$.

We cannot initialize the network to a non-binary state. Therefore, the columns of $\boldsymbol{B} = \mathrm{sgn}(\boldsymbol{B}_p)$ are our candidate patterns. To test if they are fixed points, consider

$$\mathrm{sgn}(\boldsymbol{J}\boldsymbol{B}) = \mathrm{sgn}(\boldsymbol{J}\boldsymbol{B}_p - \boldsymbol{J}\boldsymbol{E}) = \mathrm{sgn}(\boldsymbol{B}_p - \boldsymbol{J}\boldsymbol{E}).$$

We need the error $\boldsymbol{E}$ to be such that $\boldsymbol{J}\boldsymbol{E}$ will not alter the sign of $\boldsymbol{B}_p$. Two sufficient conditions are:

(a) small error: $|(\boldsymbol{J}\boldsymbol{E})_{i\mu}| < |(\boldsymbol{B}_p)_{i\mu}|$, or

(b) error with compatible sign: $(\boldsymbol{J}\boldsymbol{E})_{i\mu}(\boldsymbol{B}_p)_{i\mu} < 0$.

When either of these conditions hold for each element, we have $\mathrm{sgn}(\boldsymbol{J}\boldsymbol{B}) = \mathrm{sgn}(\boldsymbol{B}_p) = \boldsymbol{B}$, so that the candidate patterns $\boldsymbol{B}$ are fixed points. It remains to show that they are also *stable* (i.e. minima).

**Case 2:** If $\boldsymbol{W}$ is not orthogonal but its singular values remain close to one, then Löwdin Orthogonalization (also known as Symmetric Orthogonalization) (Löwdin, 1970) provides a way to preserve the HN mapping from Case 1 above.

Consider the SVD of $\boldsymbol{W}$: $\boldsymbol{W} = \boldsymbol{U}\boldsymbol{\Sigma}\boldsymbol{V}^T$. The closest matrix to $\boldsymbol{W}$ (in terms of Frobenius norm) with orthogonal columns is $\boldsymbol{L} = \boldsymbol{U}\boldsymbol{V}^T$, and the approximation $\boldsymbol{W} \approx \boldsymbol{L}$ is called the Löwdin Orthogonalization of $\boldsymbol{W}$. Note the approximation becomes exact when all the singular values are one. We then write $\boldsymbol{W}\boldsymbol{W}^T \approx \boldsymbol{U}\boldsymbol{U}^T$, and the orthogonal $\boldsymbol{W}$ approach of Case 1 can then be applied.

On the other hand, $\boldsymbol{W}$ may be strongly not orthogonal (singular values far from one). If it is still full rank, then its pseudo-inverse $\boldsymbol{W}^\dagger = (\boldsymbol{W}^T\boldsymbol{W})^{-1}\boldsymbol{W}^T = \boldsymbol{V}\boldsymbol{\Sigma}^{-1}\boldsymbol{U}^T$ is well-defined. Repeating the steps from the orthogonal case, we note here $\boldsymbol{X}^T\boldsymbol{X} = \boldsymbol{B}_p^T(\boldsymbol{W}^\dagger)^T\boldsymbol{W}^\dagger\boldsymbol{B}_p$. Defining $\boldsymbol{C} \equiv (\boldsymbol{W}^\dagger)^T\boldsymbol{W}^\dagger = \boldsymbol{U}\boldsymbol{\Sigma}^{-2}\boldsymbol{U}^T$, we arrive at the corresponding result,

$$\boldsymbol{J} = \boldsymbol{B}_p(\boldsymbol{B}_p^T\boldsymbol{C}\boldsymbol{B}_p)^{-1}\boldsymbol{B}_p^T. \tag{D.5}$$

This is analogous to the projection rule but with a "correction factor" $\boldsymbol{C}$. However, it is not immediately clear how $\boldsymbol{C}$ affects pattern storage. Given the resemblance between $\boldsymbol{J}_{\mathrm{Proj}}$ and Eq. (D.4) (relative to Eq. (D.5)), we expect that RBMs trained with an orthogonality constraint on the weights may be more readily mapped to HNs.

## D.3 EXAMPLE OF THE APPROXIMATE REVERSE MAPPING

In the main text we introduced the approximate binarization problem Eq. (12), the solutions of which provide approximately binary patterns through Eq. (13). To numerically solve Eq. (12) and obtain a candidate solution $\boldsymbol{X}^*$, we perform gradient descent on a differentiable variant.

Specifically, we replace $\mathrm{sgn}(u)$ with $\tanh(\alpha u)$ for large $\alpha$. Define $\boldsymbol{E} = \boldsymbol{W}\boldsymbol{X} - \tanh(\alpha\boldsymbol{W}\boldsymbol{X})$ as in the main text. Then the derivative of the "softened" Eq. (12) with respect to $\boldsymbol{X}$ is

$$\boldsymbol{G}(\boldsymbol{X}) = 2\boldsymbol{W}^T(\boldsymbol{E} - \alpha\boldsymbol{E} \odot \mathrm{sech}^2(\boldsymbol{W}\boldsymbol{X}))). \tag{D.6}$$

Given an initial condition $\boldsymbol{X}_0$, we apply the update rule

$$\boldsymbol{X}_{t+1} = \boldsymbol{X}_t - \gamma\boldsymbol{G}(\boldsymbol{X}_t) \tag{D.7}$$

until convergence to a local minimum $\boldsymbol{X}^*$.

In the absence of prior information, we consider randomly initialized $\boldsymbol{X}_0$. Our preliminary experiments using Eq. (D.7) to binarize arbitrary RBM weights $\boldsymbol{W}$ have generally led to high binarization error. This is due in part to the difficulty in choosing a good initial condition $\boldsymbol{X}_0$, which will be explored in future work.

To avoid this issue, we consider the case of a Hopfield-initialized RBM following CD-$k$ training. At epoch 0, we in fact have the exact binarization solution $\boldsymbol{X}^* = \boldsymbol{R}$ (from the QR decomposition, Eq. (2)), which recovers the encoded binary patterns $\boldsymbol{\xi}$. We may use $\boldsymbol{X}_0 = \boldsymbol{R}$, as an informed initial condition to Eq. (D.7), to approximately binarize the weight at later epochs and monitor how the learned patterns change.

In Fig. D.1, we give an example of this approximate reverse mapping for a Hopfield-initialized RBM following generative training (Fig. 4). Fig. D.1(a) shows the $p = 10$ encoded binary patterns, denoted below by $\boldsymbol{\xi}_0$, and Fig. D.1(b) shows the approximate reverse mapping applied to the RBM weights at epoch 10. We denote these nearly binary patterns by $\boldsymbol{\xi}_{10}$. Interestingly, some of the non-binary regions in Fig. D.1(b) coincide with features that distinguish the respective pattern. For example, the strongly "off" area in the top-right of the "six" pattern.

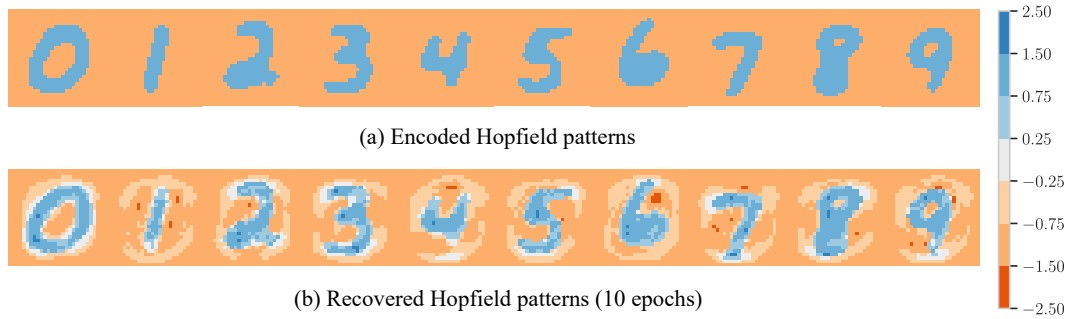

(a) Encoded Hopfield patterns

(b) Recovered Hopfield patterns (10 epochs)

Figure D.1: Reverse mapping example. (a) The $p = 10$ encoded binary patterns used to initialize the RBM in Fig. 4(a). (b) The approximate reverse mapping applied to the RBM weights after 10 epochs of CD-$k$ training. Parameters: $\alpha = 200$ and $\gamma = 0.05$.

We also considered the associative memory performance of the projection HNs built from the the patterns $\boldsymbol{\xi}_0$, $\boldsymbol{\xi}_{10}$ from Fig. D.1. Specifically, we test the extent to which images from the MNIST Test dataset are attracted to the correct patterns. Ideally, each pattern should attract all test images of the corresponding class. The results are displayed in Fig. D.2 and elaborated in the caption.

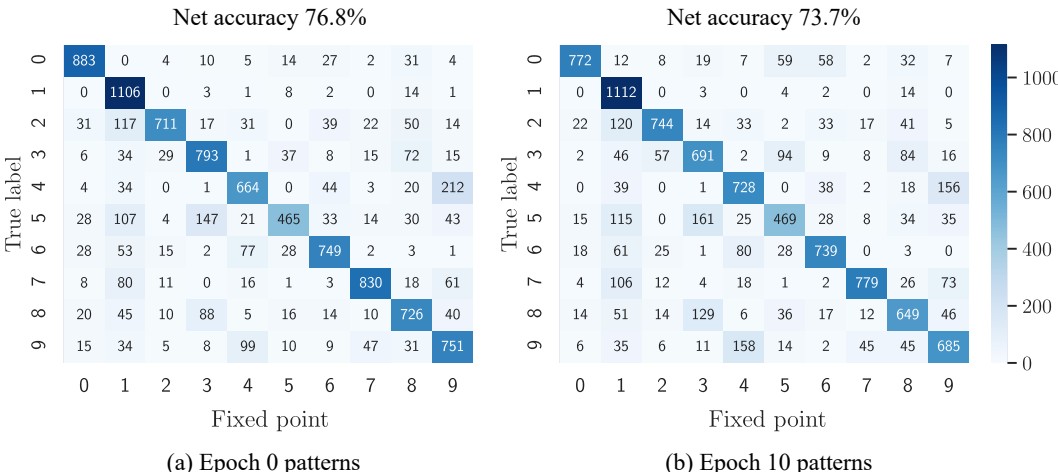

(a) Epoch 0 patterns

(b) Epoch 10 patterns

Figure D.2: Associative memory task using (a) $\boldsymbol{\xi}_0$, the initial Hopfield patterns, and (b) $\boldsymbol{\xi}_{10}$, the patterns recovered from the reverse mapping after 10 epochs. The patterns are used to construct a projection HN as in Eq. (1). Each sample from the MNIST Test set is updated deterministically until it reaches a local minimum of the HN. If the fixed point is one of the encoded patterns, the sample contributes value 1 to the table. Otherwise, we perform stochastic updates with $\beta = 2.0$ and ensemble size $n = 20$ until one of the encoded patterns is reached, defined as an overlap $N^{-1}\boldsymbol{s}^T\text{sgn}(\boldsymbol{\xi}^\mu) > 0.7$, with each trajectory contributing $1/n$ to the table.

The results in Fig. D.2 suggest that $p = 10$ patterns may be too crude for the associative memory network to be used as an accurate MNIST classifier (as compared to e.g. Fig. 5). Notably, the HN constructed from $\boldsymbol{\xi}_{10}$ performs about 3% worse than the HN constructed from $\boldsymbol{\xi}_0$, although this performance might improve with a more sophisticated method for the associative memory task. There may therefore be a cost, in terms of associative memory performance, to increasing the generative functionality of such models (Fig. 4). Our results from Appendix D.2 indicate that incorporating an orthogonality constraint on the weights during CD-$k$ generative training may provide a way to preserve or increase the associative memory functionality. This will be explored in future work.

## E   RBM TRAINING

Consider a general binary-gaussian RBM with $N$ visible and $p$ hidden units. The energy function is

$$H(\boldsymbol{s}, \boldsymbol{\lambda}) = \frac{1}{2} \sum_{\mu} (\lambda_{\mu} - c_{\mu})^2 - \sum_{i} b_i s_i - \sum_{\mu} \sum_{i} W_{i\mu} s_i \lambda_{\mu}. \tag{E.1}$$

First, we note the Gibbs-block update distributions for sampling one layer of a binary-gaussian RBM given the other (see e.g. Melchior et al. (2017)),

*Visible units*: $p(s_i = 1|\boldsymbol{\lambda}) = \frac{1}{1 + e^{-2\beta x_i}}$, where $x_i \equiv \sum_{\mu} W_{i\mu} \lambda_{\mu} + b_i$ defines input to $s_i$,
*Hidden units*: $p(\lambda_{\mu} = \lambda|\boldsymbol{s}) \sim \mathcal{N}(h_{\mu}, \beta^{-1})$, where $h_{\mu} \equiv \sum_{i} W_{i\mu} s_i + c_{\mu}$ defines the input to $\lambda_{\mu}$.

### E.1   GENERATIVE TRAINING

For completeness, we re-derive the binary-gaussian RBM weight updates, along the lines of Melchior et al. (2017). We want to maximize $L \equiv \frac{1}{M} \sum_a \ln p_{\boldsymbol{\theta}}(\boldsymbol{s}_a)$. The contribution for a single datapoint $\boldsymbol{s}_a$ has the form $\ln p_{\boldsymbol{\theta}}(\boldsymbol{s}_a) = \ln(C^{-1} \int e^{-\beta H(\boldsymbol{s}_a, \boldsymbol{\lambda})} d\boldsymbol{\lambda}) - \ln Z$ with $C \equiv (2\pi/\beta)^{p/2}$. The gradient with respect to the model is

$$\frac{\partial \ln p_{\boldsymbol{\theta}}(\boldsymbol{s}_a)}{\partial \boldsymbol{\theta}} = \frac{\int (-\beta \frac{\partial H(\boldsymbol{s}_a, \boldsymbol{\lambda})}{\partial \boldsymbol{\theta}}) e^{-\beta H(\boldsymbol{s}_a, \boldsymbol{\lambda})} \prod_{\mu} d\lambda_{\mu}}{\int e^{-\beta H(\boldsymbol{s}_a, \boldsymbol{\lambda})} \prod_{\mu} d\lambda_{\mu}} - \beta \frac{\sum_{\{s\}} \int s_i \lambda_{\mu} e^{-\beta H(\boldsymbol{s}_a, \boldsymbol{\lambda})} \prod_{\mu} d\lambda_{\mu}}{Z} \tag{E.2}$$

We are focused on the interlayer weights, with $\frac{\partial H(\boldsymbol{s}, \boldsymbol{\lambda})}{\partial W_{i\mu}} = -s_i \lambda_{\mu}$, so

$$\frac{\partial \ln p_{\boldsymbol{\theta}}(\boldsymbol{s}_a)}{\partial \boldsymbol{\theta}} = \beta (\boldsymbol{s}_a)_i \frac{\int (\lambda_{\mu} e^{-\beta H(\boldsymbol{s}_a, \boldsymbol{\lambda})} \prod_{\mu} d\lambda_{\mu}}{\int e^{-\beta H(\boldsymbol{s}_a, \boldsymbol{\lambda})} \prod_{\mu} d\lambda_{\mu}} \quad - \beta \frac{\sum_{\{s\}} \int s_i \lambda_{\mu} e^{-\beta H(\boldsymbol{s}_a, \boldsymbol{\lambda})} \prod_{\mu} d\lambda_{\mu}}{Z}$$

$$= \beta (\boldsymbol{s}_a)_i \langle \lambda_{\mu} | \boldsymbol{s} = \boldsymbol{s}_a \rangle_{\text{model}} \quad - \beta \langle s_i \lambda_{\mu} \rangle_{\text{model}}. \tag{E.3}$$

The first term is straightforward to compute: $\langle \lambda_{\mu} | \boldsymbol{s} = \boldsymbol{s}_a \rangle_{\text{model}} = \sum_i W_{i\mu} (\boldsymbol{s}_a)_i + c_{\mu}$. The second term is intractable and needs to be approximated. We use contrastive divergence (Carreira-Perpinan & Hinton, 2005; Hinton, 2012): $\langle s_i \lambda_{\mu} \rangle_{\text{model}} \approx s_i^{(k)} \lambda_{\mu}^{(k)}$. Here $k$ denotes CD-$k$ – $k$ steps of Gibbs-block updates (introduced above) – from which $s_i^{(k)}, \lambda_{\mu}^{(k)}$ comprise the final state. We evaluate both terms over mini-batches of the training data to arrive at the weight update rule Eq. (15).

### E.2   GENERATIVE PERFORMANCE

We are interested in estimating the objective function $L \equiv \frac{1}{M} \sum_a \ln p_{\boldsymbol{\theta}}(\boldsymbol{s}_a)$ during training. As above, we split $L$ into two terms,

$$L = \ln \left( C^{-1} \int e^{-\beta H(\boldsymbol{s}_a, \boldsymbol{\lambda})} \prod_{\mu} d\lambda_{\mu} \right) - \ln Z, \tag{E.4}$$

with $C \equiv (2\pi/\beta)^{p/2}$. The first term evaluates to

$$\frac{\beta}{M} \sum_{a=1}^{M} (\boldsymbol{b}^T \boldsymbol{s}_a + \frac{1}{2} ||\boldsymbol{c} + \boldsymbol{W}^T \boldsymbol{s}_a||^2) - \frac{\beta}{2} ||\boldsymbol{c}||^2, \tag{E.5}$$

which can be computed deterministically. $\ln Z$, on the other hand, needs to be estimated. For this we perform Annealed Importance Sampling (AIS) (Neal, 2001) on its continuous representation

$$Z = C^{-1}2^N \int e^{-\frac{\beta}{2}\sum_\mu(\lambda_\mu - c_\mu)^2 + \sum_i \ln\cosh\left(\beta(\sum_\mu W_{i\mu}\lambda_\mu + b_i)\right)} \prod_\mu d\lambda_\mu. \quad \text{(E.6)}$$

For AIS we need to specify the target distribution's un-normalized log-probabilities

$$\ln(Zp(\boldsymbol{\lambda})) = N\ln 2 - \frac{p}{2}\ln\left(\frac{2\pi}{\beta}\right) - \frac{\beta}{2}\sum_\mu(\lambda_\mu - c_\mu)^2 + \sum_i \ln\cosh\left(\beta\left(\sum_\mu W_{i\mu}\lambda_\mu + b_i\right)\right), \quad \text{(E.7)}$$

as well as an initial proposal distribution to anneal from, which we fix as a $p$-dimensional unit normal distribution $\mathcal{N}(\mathbf{0}, \boldsymbol{I})$.

### E.3 CLASSIFICATION PERFORMANCE

Here we provide extended data (Fig. E.1) on the classification performance shown in Fig. 5. As in the main text, color denotes initial condition (introduced in Section 3.1) and shape denotes the number of sub-patterns. Here we train for each 100 epochs, which allows convergence of most of the curves. We also include the "Hebbian" initialization (green curves).

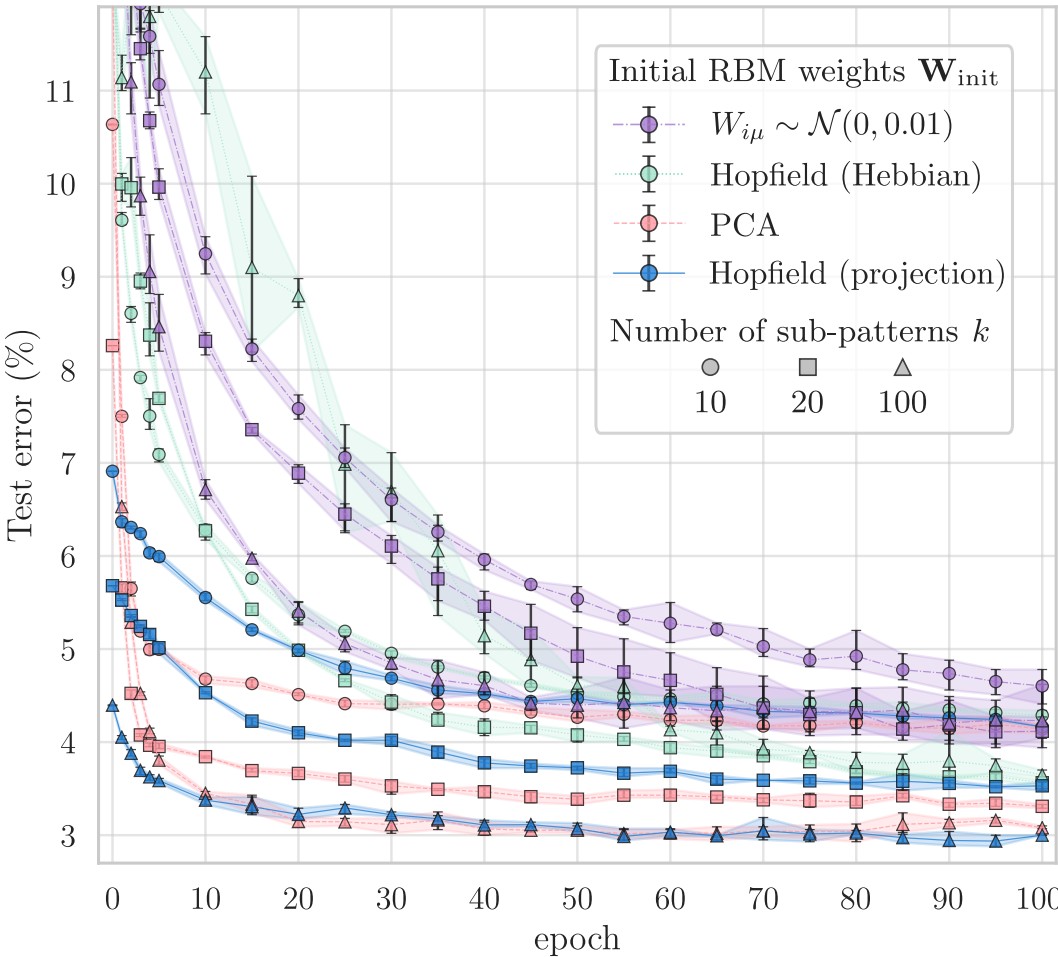

Figure E.1: Product-of-experts classification performance (extended data).

Notably, the Hebbian initialization performs quite poorly in the classification task (as compared to direct generative objective, Fig. 4). In particular, for the 100 sub-patterns case, where the projection

HN performs best, the Hebbian curve trains very slowly (still not converged after 100 epochs) and lags behind even the 10 sub-pattern Hebbian curve for most of the training. This emphasizes the benefits of the projection rule HN over the Hebbian HN when the data is composed of correlated patterns, which applies to most real-world datasets.

