# OpenReview forum: "On the mapping between Hopfield networks and Restricted Boltzmann Machines"
_ICLR.cc/2021/Conference — ICLR 2021 Oral_

### Official Review · AnonReviewer3 · 2020-10-25
**A theoretical link between BN and RBM, but experiments are worth improving**

**Rating:** 7
**Confidence:** 4

**Review:**

# Summary

This paper shows a relationship between the project rule weights of a Hopfield network (HN) and the interaction weights in a corresponding restricted Boltzmann machine (RBM). The mapping from HN to RBM is facilitated by realising that the partition function of BN can be seen as the partition function of a binary-continuous (Bernoulli-Gaussian) RBM. The authors comments on the mapping from RBM to BN. The experiments show the advantages of training RBM with weights initialised from BN projection weights in generation and classification.

## Strong points:
+ I am not familiar with the literature, but the results seem new to me.
+ The experiments show advantages of BN initialisation, pointing to new directions of improving RBM training.
+ The paper is fairly clearly written.

## Weak points

- The HN -> RBM mapping is quite clear, but the reverse RBM -> HN mapping is not very well established, and there are no experiments showing how effective the approximate reverse mapping works on associative memory tasks typical for HNs. I also believe this lowers the impact of this paper, given that the forward mapping is based on a simple revelation.
- The authors' description of the experimental results are not accurate enough. and the results raise several questions to be addressed.

# Recommendation

I'm in favour of rejection, but some concerns can be addressed fairly easily (with experiments) so I'm open to raising my score if questions are well-addressed.

## Issues and questions to address

* The authors should provide experiments on the reverse mapping as suggested above.
* I do not agree that figure 3 shows that RBM training "simply 'fine tunes'" the weights -- the difference is quite stark. How about increasing the batch size so that there is little SGD noise?
* Figure 4a: traces are cut-off just when random initialization is catching up with HN initialization. This also applies to Figure 5.
* There are a few descriptions suggesting "HN init. appears to train much faster than random init". However, the rate of increase in of likelihood in Figure 4 is shallower for HN than for rand init. Is the advantage only at the 0'th RBM epoch?
* The author only compared with purely random initialisation, which is perhaps the most naive baseline. I would suggest comparing to a (slightly) more clever initialisation, perhaps PCA or something better (those mappings in previous work the authors cited and in Appendix B). Or, the authors could also initialise the RBM by first training it on the within-class cluster centres (using a very large number of sleep samples for the sleep-phase) which may also be a more fair comparison?
* In the classification objective, if I understand correctly, the feature function is essentially quadratic in the input patterns. Should there be an ideal test error that is computed by a quadratic neural network trained with supervision by backpropagation? If the HN classifier (blue in Figure 5) is approaching this idealization, then this will strengthen the claim.
* The discussion on the extension to more generic, deep architectures is not well supported, and I do not see the extension to be so straightforward given the content of the current paper. In generation, supervised labels and clustering are used to simplify learning. Is the network able to learn just on the MNIST digits, even for the real images within a single class (e.g. "7")?
* Can the authors try to characterise whether the HN initialisation is related to log-likelihood training? I wonder if there is any interesting theory; otherwise, measuring model performance by log-likelihood seems a bit arbitrary (though it makes the comparison to contrastive divergence easier).

# Detailed suggestions (not to affect decision)

* Reference to RBMs should include more historic ones from Hinton (e.g. 2006)
* I do not see the purpose of (7) and (8), and they are only referred to in the Appendix (the review content in the Appendix is informative by itself though).
* Eqn (11), should it be $w_\mu w_\mu^T$ in the sum?
* Third line above (16), should $H$ and $Z$ be indexed by $\mu$?
* Line above (D.4), $WW^T = \dots B_p^T$?


==== update ====

I thank the authors for providing such detailed response. All my concerns are addressed and reflected in the revision (though some are much better done than the rest). I congratulate the authors on their spirit of maintaining a high standard on the theory, experiments and descriptions, and therefore significantly raise my score. I hope to see this paper accepted.

---

> ### Author Response · Authors · 2020-11-16
> **Response to Reviewer 3 (pt 2/2)**
>
> (6) *"In the classification objective, if I understand correctly, the feature function is essentially quadratic in the input patterns. Should there be an ideal test error that is computed by a quadratic neural network trained with supervision by backpropagation? If the HN classifier (blue in Figure 5) is approaching this idealization, then this will strengthen the claim."*
> The reviewer is correct that the feature function is quadratic in the input states (each feature requires a $N \times k$ matrix of weights $W_{ik}^{\mu}$ ). We are not familiar with theoretical results/bounds on feedforward neural network classification performance when using analogous quadratic layers. We note that the fact that both PCA and HN for 100 sub-patterns converge to the same limit after many epochs indicates the possibility that they may be reaching such a bound, although we are not able to prove it at this point. We have updated the text accordingly.
>
> (7a) *"The discussion on the extension to more generic, deep architectures is not well supported, and I do not see the extension to be so straightforward given the content of the current paper."*
> We agree that to establish the feasibility of such mappings will require future research and we have revised the text accordingly. Further, to substantiate this suggestion we have added an example to Appendix C (see Eq. (C.5)) showing one way that higher-than-2 layer networks can arise from these ideas.
>
> (7b) *"In generation, supervised labels and clustering are used to simplify learning. Is the network able to learn just on the MNIST digits, even for the real images within a single class (e.g. "7")?"*
> We are unsure if this is related to the reviewer's previous bullet point (we split them). The hopfield network, and associated mapping, relies on having access to "patterns". These patterns can be the centroids of labelled clusters, for example. The clustering does not have to be supervised. In that sense it can learn on just the MNIST digits from a single class, which is roughly what is done in the classification section.
>
> (8) *"Can the authors try to characterise whether the HN initialisation is related to log-likelihood training? I wonder if there is any interesting theory; otherwise, measuring model performance by log-likelihood seems a bit arbitrary (though it makes the comparison to contrastive divergence easier)."*
> Maximizing the log-likelihood of the data is equivalent to minimizing the KL divergence between the model distribution and the data distribution (a standard generative objective). This is why we display it in Fig. 4.
>
> The intuitive reason for why the HN initialization works well is because the Hopfield patterns capture the key features/prototypes from the dataset. The projection rule encodes the patterns (and nearby states) as high probability basins in the free energy landscape. Because the data itself is clustered near the patterns, these basins model the true data distribution well, which is reflected in the good generative performance without training. We hope this addresses the reviewer's question and are happy to discuss further.
>
> *"Detailed suggestions (not to affect decision):"*
> In addition to the points above, we have also incorporated the reviewer's detailed suggestions into the manuscript. Namely, additional references to Hinton's early work on RBMs, further detail near Eq. (7), and correction of the indicated typos.
>
> We thank the reviewer again for their excellent suggestions. We hope that these changes address their concerns.

---

> ### Author Response · Authors · 2020-11-16
> **Response to Reviewer 3 (pt 1/2)**
>
> We thank the reviewer for their careful reading and constructive feedback towards improving our manuscript. We address the reviewer’s points below (numbered in order) and have updated our manuscript accordingly:
>
> (1) *"The authors should provide experiments on the reverse mapping as suggested above."*
> This is an excellent point. We agree with the reviewer that better understanding the reverse mapping is an important next step. We see as one of the important applications of the reverse mapping its potential to provide insight into what classes the RBM has “learned” after training. As our preliminary results suggest (Appendix D.2), the reverse mapping is most likely to be feasible when the RBM weights are approximately orthogonal. Thus, a prerequisite step would be to incorporate an orthogonality constraint to the weight updates during CD-k training. However, given the time constraints, we feel that the full investigation is beyond the scope and the focus of the current manuscript. We hope this addresses the reviewer’s point.
>
> (2) *"I do not agree that figure 3 shows that RBM training "simply 'fine tunes'" the weights -- the difference is quite stark. How about increasing the batch size so that there is little SGD noise?"*
> We thank the reviewer for pointing this out. We have increased the batch size to 1000. With this batch size, the weights after 50 epochs are now significantly closer to the HN initialization than with the previous lower batch size. This emphasizes the fact the HN initialization performs extremely well without any or with very little amount of training (see also responses to points (4) and (5) below). We have accordingly updated Fig. 3, Fig. 4a (to show convergence with the larger batch size), and the wording above Fig. 3.
>
> (3) *"Figure 4a: traces are cut-off just when random initialization is catching up with HN initialization. This also applies to Figure 5."*
> We have extended Fig. 4a to 60 epochs to show that the random initialization converges to the same value as HN initialization. We have included in Section E.3 an extended version of Fig. 5 (including extra initial conditions, see point (5)) with training to 100 epochs to better show convergence.
>
> (4)*"There are a few descriptions suggesting "HN init. appears to train much faster than random init". However, the rate of increase in of likelihood in Figure 4 is shallower for HN than for rand init. Is the advantage only at the 0'th RBM epoch?"*
> We apologize for the confusing phrasing. By faster we mean that it is closer to its peak value after a smaller number of epochs. Indeed, HN initialization converges fastest within the 0’th epoch, and after that the rate of convergence is slower (because HN initialization has almost reached the limit). We have re-phrased the appropriate parts of the manuscript more precisely. This also emphasizes the fact that HN initialization performs very well already with very limited training (see also response to points (2) and (5)).
>
> (5) *"The author only compared with purely random initialisation..."*
> This is an excellent point. We had initially focused on the random initialization because it commonly used. To address this concern, at the reviewer's suggestion, we have performed additional experiments with two alternative initializations: (1) PCA and (2) the "Hebbian" Hopfield mapping developed in previous work for uncorrelated patterns (our mapping uses the “projection” Hopfield Network, denoted “HN” below). We have updated Fig. 4, Fig. 5, and the text accordingly.
>
> Interestingly, in Fig. 4 although all four initializations eventually converge to the same limit, all three – HN, PCA and “Hebbian” initialization perform significantly better than the random initialization after relatively limited amount of training. Importantly, HN initialization outperforms both PCA and Hebbian initialization at early times – emphasizing the fact that HN initialization performs well with zero CD-k training. For the classification objective (Fig. 5), the advantage of HN relative to PCA and Hebbian is more pronounced: for instance, for 100 sub-patterns the PCA becomes comparable to HN only after ~5 epochs, and further emphasizes the performance of HN with zero training. For classification, Hebbian is significantly lagging behind (projection) HN, which is now shown in the appendix Fig. E.1 (along with longer training time, see point (3)).
>
> This could be of potentially applied importance for rapid training of very large datasets, but we wish to emphasize that the main theoretical point of the paper is that we provide a novel mapping between two classical models, which as an added benefit provides a reasonable and potentially useful RBM initialization. Furthermore, future work focusing on the differences in learning during the first few epochs (among the various initializations) may provide insights into what is actually being “learned” by the RBM during this time.

---

> ### Author Response · Authors · 2020-11-23
> **Additional response to Reviewer 3**
>
> Please note that we have uploaded an updated version of the revised manuscript.
>
> To further address point (1) of Reviewer 3 (*"The authors should provide experiments on the reverse mapping as suggested above."*), Appendix D.3 now contains an example of the approximate reverse mapping along with an example of the performance on an associative memory task.
>
> We hope this addresses the reviewer's concerns.

---

### Official Review · AnonReviewer4 · 2020-10-26
**ICLR review for "On the mapping between Hopfield networks and Restricted Boltzmann Machines"**

**Rating:** 7
**Confidence:** 4

**Review:**

This paper considers a mapping between the well known Hopfield Neural Networks and Restricted Boltzmann Machines. In contrast with previous literature that consider the case where the patterns / data features to memorize were uncorrelated, the authors extend the mapping to arbitrarily correlated patterns, which allows to consider much more realistic settings. The mapping is computationally speaking relatively cheap. This mapping is shown to allow for significantly better initialization (than random) of the weights of a RBM, in the sense that the training is then much faster to reach comparable generative and/or generalization performance. In this sense the mapping is not only interesting from a theoretical point of view, but also practically. This paper should be considered as an applied one, as there is no real analytic theory of why this mapping helps the learning, but the experiments are well carried: the boost in learning is demonstrated through experiments in MNIST data, and the results are well explained and convincing. The appendices are also well written and are a good addition to the main part. Overall the paper is well written (the paper can be used by non-specialists also as introduction to Hopfield NNs and RBMs), the results are interesting and relevant to the ML community, the paper can be read without much effort. Even if RBM are not anymore state-ot-the art generative models, the results are encouraging and might lead to future improvements in more modern architectures. I have no specific concern. The paper is overall very well written. The paper is slightly incremental as similar mappings were known, but it remains a relevant contribution, and the aspect of using this mapping as a way to boost learning in RBM seems new, and interesting. I recommend publication after slight corrections, see below.

Typos and corrections:
_text below (1): J=1/N Xi^T Xi^T ->1/N Xi Xi^T
_(3): please detail the last equality
_(8) is true only for the lambda that verify the fixed point / saddle point equations: please mention it
_below (11): the p the columns of -> the p columns of
_(12): please explain what is GL_p(R)
_"At the other end, 0 ≪ 10k/N < 1, ..." : Any x > 0 is >> 0, so please be more precise
_Above Fig 3: "appear qualitatively similar" : this is not obvious...

---

> ### Author Response · Authors · 2020-11-16
> **Response to Reviewer 4**
>
> We thank the reviewer for their detailed review of our manuscript and positive feedback. In the updated version of our manuscript, we have corrected the noted typos and adjusted the text near Eq. (3), (8), and (12). We have also clarified the comments on the limited capacity of stored sub-patterns ($10k/N < 1$), and on the qualitative similarity between Fig. 3a and Fig. 3b (we updated the figure after training with a larger batch size as suggested by another reviewer).

---

### Official Review · AnonReviewer1 · 2020-10-31
**A nice theoretical exposition and result!**

**Rating:** 10
**Confidence:** 4

**Review:**

The paper demonstrates a mathematical equivalence between Hopfield nets and RBMs, and it shows how this connection can be leveraged for better training of RBMs.

What a great paper - well written, an enlightening mathematical connection between two well-known models that to my knowledge was not previously known.  Hopfield nets and RBM's have been around for decades, and I don't think we've been aware of this connection, so it seems like a pretty important finding.  The paper explores the utility of this connection by applying to an MNIST task.  Interestingly, the connection yields important insights in both directions: stochastic sampling in an RBM is faster than Hopfield due to a smaller matrix and parallel layer wise updates, whereas initializing an RBM with the projection rule from Hopfield allows it to find a better solution faster.

I really enjoyed reading the paper, I learned something new, and I think others will too!  It is an important advance in our understanding of Hopfield nets and RBMs.

---

> ### Author Response · Authors · 2020-11-16
> **Response to Reviewer 1**
>
> We thank the reviewer for their time and positive feedback. Many thanks for the strong support of our work!

---

### Decision · Program_Chairs · 2021-01-07
**Final Decision**

**Decision:**

Accept (Oral)

**Comment:**

Two knowledgeable reviewers were positive 7 and very positive 10 about this paper, considering it an important contribution that illuminates previously unknown aspects of two classic models, namely RBMs and Hopfield networks. They considered the work very well developed, theoretically interesting and also of potential practical relevance. A third reviewer initially expressed some reservations in regard to the inverse map from RBMs to HNs and the experiments. Following the authors' responses, which the reviewer found detailed and informative, he/she significantly raised his/her score to 7, also emphasizing that he/she hoped to see the paper accepted. With the unanimously positive feedback, I am recommending the paper to be accepted.